# The Complete Mitochondrial Genome of *Spirobolus bungii* (Diplopoda, Spirobolidae): The First Sequence for the Genus *Spirobolus*

**DOI:** 10.3390/genes13091587

**Published:** 2022-09-03

**Authors:** Hanmei Xu, Yu Fang, Guohua Cao, Caiqin Shen, Hongyi Liu, Honghua Ruan

**Affiliations:** 1Co-Innovation Center for Sustainable Forestry in Southern China, College of Biology and the Environment, Nanjing Forestry University, Nanjing 210037, China; 2Dongtai National Forest Farm of Jiangsu Province, Dongtai 224200, China

**Keywords:** Diplopoda, mitochondrial DNA, rearrangement, transcription direction, phylogenetic tree

## Abstract

Millipedes (Diplopoda) comprise one of the most important groups of large soil arthropods in terrestrial ecosystems; however, their phylogenetic relationships are poorly understood. Herein, the mitochondrial genome (mitogenome) of *Spirobolus bungii* was sequenced and annotated, which was 14,879 bp in size and included 37 typical mitochondrial genes (13 protein-coding genes (PCGs), two ribosomal RNA genes (rRNAs), and 22 transfer RNA genes (tRNAs)). Most of the 13 PCGs had ATN (AT/A/T/G) as the start codon except for COX1, which used CGA, and most PCGs ended with the T end codon. By comparing the gene arrangements of the mitogenomes among Diplopoda species, rearrangement occurred between and within orders. In contrast to *Narceus annularus*, the mitogenome genes of *S. bungii* had consistent orders but were transcribed in completely opposite directions, which was a novel finding in Spirobolidae. Moreover, the phylogenetic relationships within Diplopoda, which were based on the sequences of 13 PCGs, showed that *S. bungii* was clustered with *N. annularus*, followed by *Abacion magmun*. This indicated that there might be a close relationship between Callipodida and Spirobolida. These results could contribute to further studies on the genetics and evolutionary processes of *S. bungii* and other Diplopoda species.

## 1. Introduction

Millipedes *Spirobolus bungii* (*S. bungii)* belongs to the Spirobolidae family of the Diplopoda class [1]. Diplopoda comprise one of the most important groups of large soil arthropods in the terrestrial ecosystems [2], with key decomposition and nutrient cycling functions in forests [3]. They also serve as model organisms for addressing myriad evolutionary, ecological, and biological concepts and questions [4]. Diplopoda are found worldwide and reside within forests, meadows, mountains, caves, farmlands, urban green spaces, and residential areas [1]. While there have been interesting studies on millipedes in recent years, they remain a largely unexplored group, with only 12,000 of the predicted 60,000 [5] to 80,000 [6] species that are currently described. To date, there are very few studies on Diplopoda and even fewer for species in China [7,8]. Furthermore, phylogenetic studies based on morphological characteristics between diplopod taxa are rare [9,10].

Molecular data have become increasingly important in recent years. In animals, the typical mitochondrial genome (mitogenome) is a circular double-stranded DNA molecule, which encodes 13 protein-coding genes (PCGs) for the enzymes required for oxidative phosphorylation, two ribosomal RNA genes (rRNAs), and 22 transfer RNA genes (tRNAs) necessary for the translation of the proteins encoded by the mitogenome [11,12]. Compared with individual genes, the mitogenome remains a promising tool for inferring phylogenetic relationships due to its high information content. Recently, some mitogenomes in Diplopoda were published and applied to explore phylogenetic relationships [13,14,15,16]. However, only a few mitogenomes have been published for Spirobolida [17]. Further, the arrangement of genes in mitogenomes is remarkably variable across Diplopoda [13,17,18].

In this study, for the first time, the *S. bungii* mitogenome was assembled and characterized. The structural organization, nucleotide composition, codon usage, and AT/GC-skew were analyzed. Additionally, we conducted phylogenetic analyses based on 13 PCGs available elsewhere for the purpose of investigating the phylogenetic position of *S. bungii* within Diplopoda, which might further elucidate the genetics and evolutionary processes of *S. bungii* and other Diplopoda species.

## 2. Materials and Methods

### 2.1. Sample Collection and DNA Extraction

The specimens used in this study were collected from the Purple Mountain (30°01′ N, 118°48′ E) in 2019, where an existing deciduous broadleaved mixed forest is dominated by oaks (e.g., *Quercus varialis* BL, *Q. accutissima* Carruth), in Nanjing, Jiangsu Province, China. Following morphological identification, the samples were stored at −20 ℃ in the Ecology Laboratory of Nanjing Forestry University (Accession No: NFU20191103). The total genomic DNA was prepared from a small portion of body segments of a single individual using the SDS-protease K-alcohol phenyl-trichlormethane method. The remaining tissue was stored at −20 °C in 90% ethanol to preserve the specimens. 

### 2.2. Mitogenome Sequencing, Assembly, and Annotation

The complete genomic library of *S. bungii* was established using an Illumina HiSeqNano DNA Sample Prep Kit (Illumina, San Diego, CA, USA), whereas the sequencing was performed using next-generation sequencing (NGS) via Illumina Hiseq2000 (Illumina, USA). To generate clean data, low-quality sequences were removed. About 40 million reads with a GC content of 43.65% were assembled to obtain a complete mitogenome using SPAdes v3.11.1 [19]. Thus, the complete mitochondrial genome sequence was used to predict the transcriptional direction of each gene component using the Improved de novo Metazoan Mitochondrial Genome Annotation (MITOS) platform [20]. The annotated mitochondrial genome sequence of *S. bungii* was submitted to GenBank (Accession: NC_056899.1). 

### 2.3. Sequence Analysis

The mitochondrial ring structure was plotted by comparative genomics (CG) View Server [21], and 22 tRNA clover two-dimensional structures were predicted using tRNAscan-Se [22]. The composition skew was calculated according to the following formulae: AT-skew = (A − T)/(A + T) and GC-skew = (G − C)/(G + C) [23]. Next, a visual graph of the composition skew was created using the ggplot2 packages in R v.4.2.0. Moreover, the R script for the relative synonymous codon usage (RSCU) frequency graph was generated from PhyloSuite [24], which was then run in R v.4.2.0.

### 2.4. Phylogenetic Analysis

To clarify the phylogenetic position of *S. bungii*, the available complete mitogenomes were obtained from GenBank and were comprised of nine orders and 27 species (Table 1). *Stylochyrus rarior* (GenBank accession: CQ927176.2) from order Mesostigmata was used as the outgroup. A total of 27 species, including *S. bungii*, were employed to develop phylogenetic trees based on 13 PCGs.

All operations were performed with the PhyloSuite software package [24]. The sequences were aligned in batches using MAFFT software [25]. Ambiguously aligned areas were removed using Gblocks [26]. ModelFinder was utilized to partition the codons and identify the best substitution model for phylogenetic analyses [27]. Phylogenetic trees were constructed with Bayesian inference (BI) and maximum likelihood (ML). The ML phylogenies were inferred using IQ-TREE [28] under the model automatically selected by IQ-TREE (‘Auto’ option in IQ-TREE) for 5000 ultrafast [29] bootstraps, as well as the Shimodaira–Hasegawa-like approximate likelihood-ratio test [30]. BI analysis was performed using MrBayes v.3.2.6 [31] with four chains (one cold chain and three hot chains). Two independent runs of 2,000,000 generations were conducted with sampling every 100 generations. The first 25% of trees were discarded as burn-in.

## 3. Results and Discussion

### 3.1. Mitogenome Structure and Organization

Akin to other well-characterized firefly mitochondrial genomes, the mitogenome of *S. bungii* was a double-stranded circular DNA molecule, which contained 37 typical mitochondrial genes (13 PCGs, 22 tRNAs, and two rRNAs) (Figure 1 and Table A1). Four PCGs (ND1, ND4L, ND4, and ND5), two rRNAs, and nine tRNAs (trnV, trnL(UAG), trnL(UAA), trnP, trnH, trnF, trnY, trnQ, and trnC) were transcribed from the major stand (J-stand), and the other genes from the minor strand (N-strand) (Figure 1 and Table 2). Fifteen intergenic spacers were observed between the mitochondrial regions with lengths between −6 and 40 bp. Among these intergenic spacers, the longest was 17 bp (found between trnQ and trnT) (Table 2).

The complete mitochondrial genome was 14,879 bp in size, and its overall base composition was 26.60% for A, 32.62% for T, 28.44% for G, and 12.34% for C, with a GC content of 40.78% (Table 3), which was slightly higher than other Diplopoda species (Table A1) [15,32]. The AT-skew of *S. bungii* was negative, while the GC-skew was positive, which was opposed to *Narceus. annularus* in the same family Spirobolidae (Figure 2). Further, the GC-skews of all Polydesmida species were positive, while the AT-skews for all of this order were negative, which was completely opposed to the Spirostretida order (Figure 2).

### 3.2. The PCGs

The total length of the PCGs was 10,977 bp, which was consistent with other Diplopoda species (Table A1). The base composition of the PCGs was A = 24.53%, T = 32.22%, G = 20.46%, and C = 22.78% (Table 3). In contrast to the whole mitochondrial genome, the AT- and GC-skews were both negative, which were the same as the almost Spirostreptida species (Figure 2).

The gene arrangements of 13 PCGs were COX1, ND2, ND1, ND4L, ND4, ND5, Cytb, ND6, ND3, COX3, ATP6, ATP8, and COX2. Half of the PCGs began with a common ATG start codon, and most PCGs ended with a T end codon (Table 2). In the 13 PCGs, ND1, ND4L, ND4, ND5, CYTB, ND6, ND3, COX3, ATP6, ATP8, COX2, and ND2 used ATN (ATA/T/G/C) as the start codon, while COX1 was initiated by CGA. All PCGs stopped with TAA/G or with their incomplete single T form (Table 2). The single T as the stop codon has been found in other species [33,34,35,36].

The RSCU of the *S. bungii* mitogenome is presented in Figure 3, which indicates that Leu, Val, and Gly were the three most frequently utilized amino acids, and Cys had the lowest concentration (Figure 3B). Nine of the twenty-two amino acids (i.e., Pro, Thr, Leu1, Arg, Ala, Ser1, Ser2, Val, and Gly) had four codons, while the others had two (Figure 3A).

### 3.3. Transfer RNAs and Ribosomal RNAs

The typical sets of the 22 tRNAs were identified with sizes ranging from 57 bp (trnS) to 68 bp (trnQ) (Table 2). Moreover, the total length of the tRNAs was 1375 bp, with an A+T content of 65.02%, an AT-skew of 0.056, and a GC-skew of 0.089. Among all secondary structures of the 22 tRNA genes from the *S. bungii* mitochondrial genome, except for trnS1, all had a typical cloverleaf structure (Figure 4), as observed in other Diplopoda mitogenomes [10].

For *S. bungii*, the rrnL gene (length: 1270 bp) was encoded between trnV and trnL1, and the rrnS gene was 803 bp long. The total size of the two rRNAs was 2073 bp, with an A+T content of 66.14%, an AT-skew of −0.102, and a GC-skew of 0.345, which were higher than the other regions (Table A1). The rRNA AT-skews of all these species were positive, while the GC-skews were negative except for *Anaulaciulus gracilipes* (Figure 2).

### 3.4. Phylogenetic Analysis

Based on ML and BI analyses of nucleotide data of the 13 PCGs, we reconstructed the phylogenetic relationships of 26 species of Diplopoda, with *S. rarior* (Arachnida) as an outgroup. The two trees were similar to each other, with strongly supported branches (Figure 5). For the BI tree, Callipodida was clustered with Sphaerotheriida and Glomeridesmida, while it did not cluster with any species for the ML tree. However, *S. bungii* was most closely related to *N. annularus*, and the relationships between Callipodida, Spirobolida, Julida, and Spirostreptida were stable, which was congruent with a previous study of mitochondrial genomes [32].

### 3.5. Gene Arrangement among Diplopoda Classes

By comparing the gene arrangements of the mitogenomes between Diplopoda species, rearrangement occurred between and within orders (Figure 6). The positions of trnT for Julida differed from those of Spirobolida and Spirostreptida, which had similar gene arrangement patterns (Figure 6). Within Julida, the positions of trnC and trnW were inversed (Figure 6), which were found in fireflies [37]. An interesting phenomenon occurred where the gene orders of the mitogenomes between *S. bungii* and *N. annularus* were consistent, while they were transcribed in completely opposite directions (Figure 6). This was also found in the Glomeridesmidae family of the Glomeridesmida order (Figure 6). Further, the positions of trnP in *Antrokoreana gracilipes* and *Anaulacilus koreanus*, which belonged to Julida order, were consistent but transcribed in opposite directions (Figure 6).

## 4. Conclusions

The mitogenome of *S. bungii* was determined to be 14,879 bp in length, with a GC content of 40.78%. Additionally, based on a mitogenomic analysis of *S. bungii*, we found an intriguing phenomenon, where the AT- and GC-skews of the *S. bungii* mitogenome were opposed to most Diplopoda, while those of the 13 PCGs were consistent, except for Polydesmida. Consequently, this mitogenome, particularly the 13 PCGs, will assist with elucidating the genetic diversity, evolutionary origins, and genetic relationships of Diplopoda. The arrangement of genes in mitogenomes was remarkably variable across Diplopoda. Conversely, the mitogenome genes had consistent orders; however, for the Glomeridesmida and Spirobolida orders, they were transcribed in opposite directions. This indicated that the phenomenon was prevalent in Diplopoda, which will warrant additional investigations in the future. Furthermore, these results provide valuable data for the future resolution of phylogenetic relationships in this tribe.

## Figures and Tables

**Figure 1 genes-13-01587-f001:**
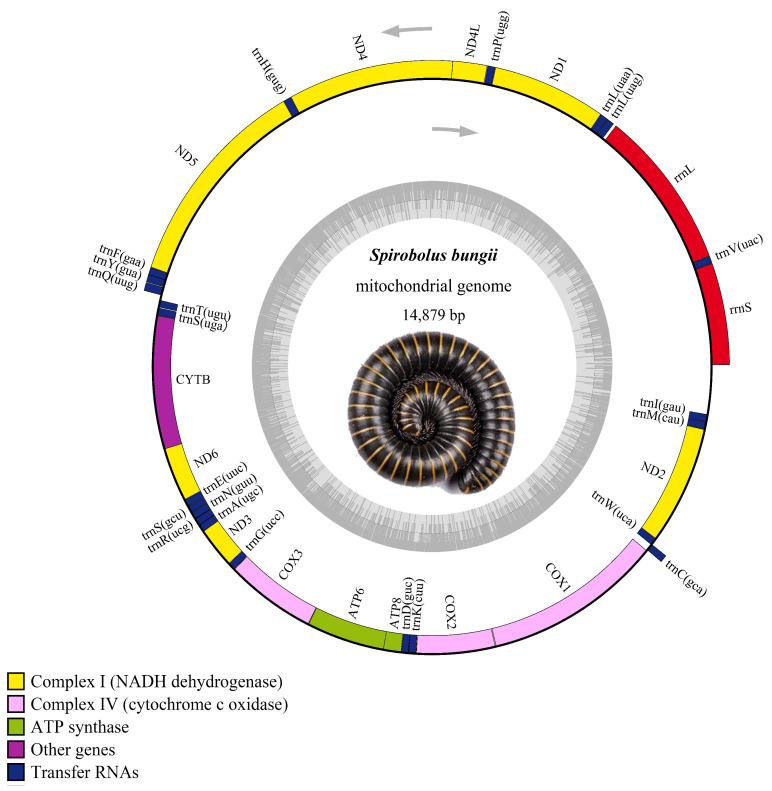
Circular map of the mitogenome of *S. bungii*. The circle shows the gene map of *S. bungii* where genes outside the map are coded on the major strand (J-strand), and those on the inside of the map are coded on the minor strand (N-strand). Genes are represented by differently colored blocks.

**Figure 2 genes-13-01587-f002:**
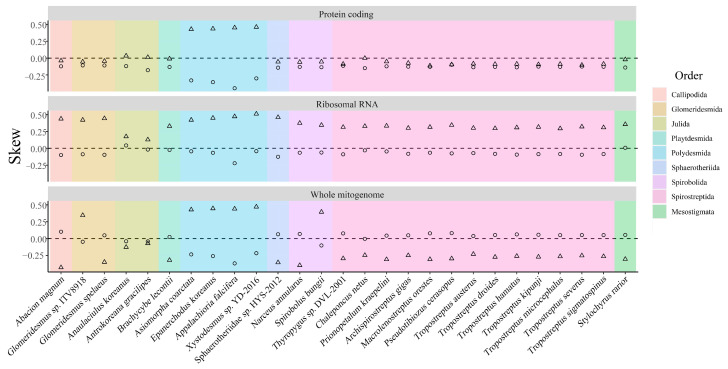
AT-skew (circle) and GC-skew (triangle) of 27 species used in this study.

**Figure 3 genes-13-01587-f003:**
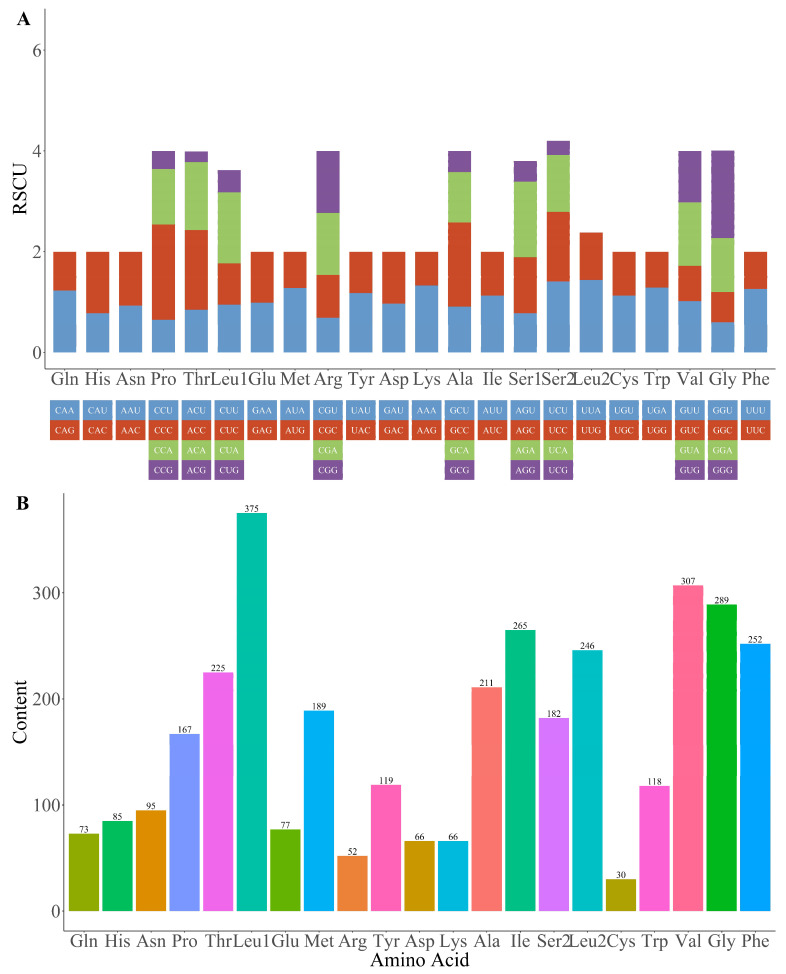
Relative synonymous codon usage (**A**) and codon distribution (**B**) in *S. bungii* mitogenome.

**Figure 4 genes-13-01587-f004:**
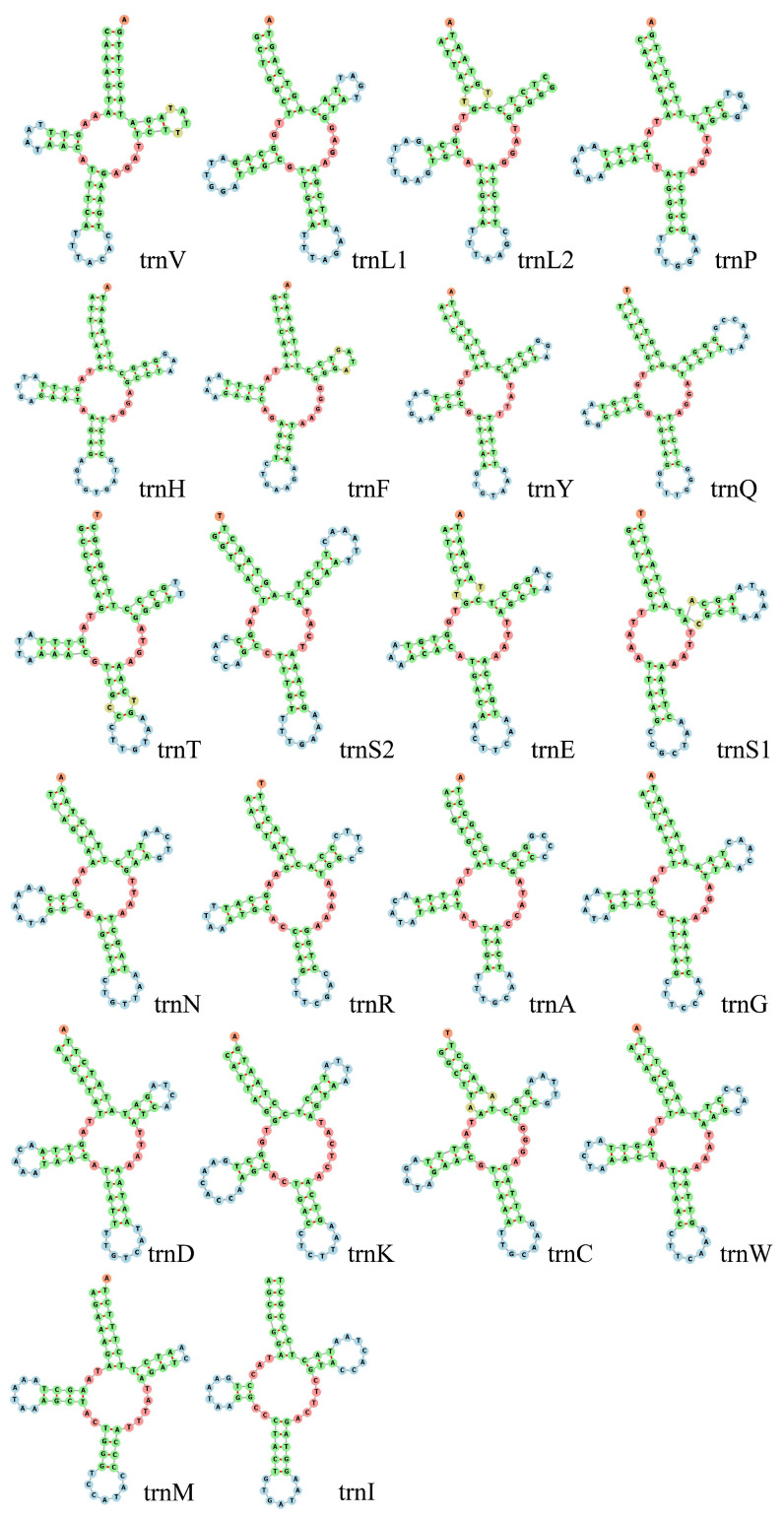
Secondary structure of 22 tRNA genes from the *S. bungii* mitochondrial genome.

**Figure 5 genes-13-01587-f005:**
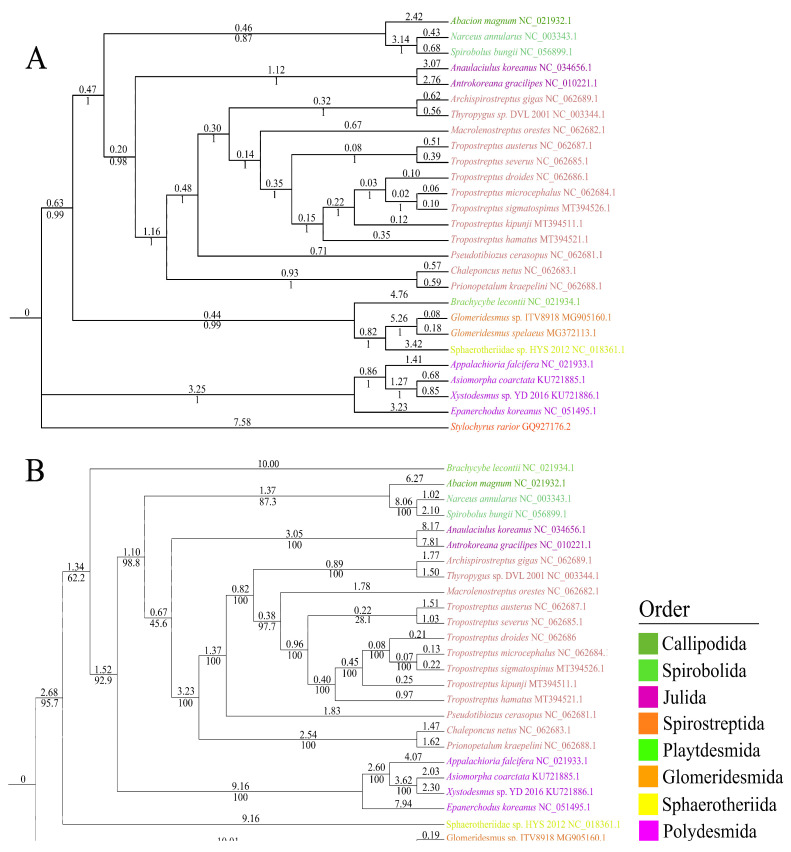
Mitogenomic phylogeny of 26 Diplopoda species and an outgroup (*Stylochyrus rarior*) based on 13 PCGs using Bayesian inference (**A**) and maximum likelihood (**B**) methods. The same colors of species in the tree indicated the same order.

**Figure 6 genes-13-01587-f006:**
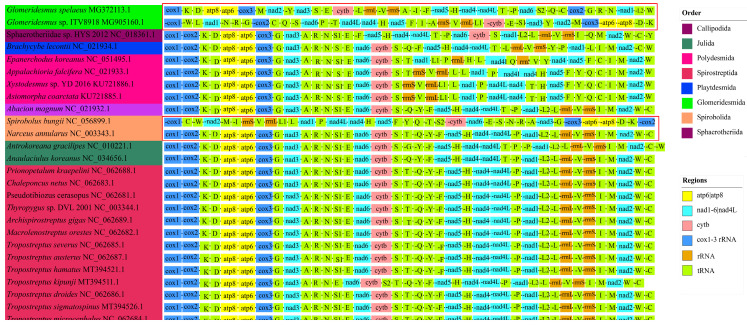
Gene arrangement image of Diplopoda mitogenomes.

**Table 1 genes-13-01587-t001:** List of complete mitogenomes used in this study.

Class	Order	Family	Genus	Species	Accession
Diplopoda	Callipodida	Callipodidae	*Abacion*	*Abacion magnum*	NC_021932.1
	Glomeridesmida	Glomeridesmidae	*Glomeridesmus*	*Glomeridesmus* sp. ITV8918	MG905160.1
				*Glomeridesmus spelaeus*	MG372113.1
	Julida	Julidae	*Anaulaciulus*	*Anaulaciulus koreanus*	NC_034656.1
		Nemasomatidae	*Antrokoreana*	*Antrokoreana gracilipes*	NC_010221.1
	Playtdesmida	Andrognathidae	*Brachycybe*	*Brachycybe lecontii*	NC_021934.1
	Polydesmida	Paradoxosomatidae	*Asiomorpha*	*Asiomorpha coarctata*	KU721885.1
		Polydesmidae	*Epanerchodus*	*Epanerchodus koreanus*	NC_051495.1
		Xystodesmidae	*Appalachioria*	*Appalachioria falcifera*	NC_021933.1
			*Xystodesmus*	*Xystodesmus* sp. YD-2016	KU721886.1
	Sphaerotheriida	Sphaerotheriidae	*N/A*	*Sphaerotheriidae* sp. HYS-2012	NC_018361.1
	Spirobolida	Spirobolidae	*Narceus*	*Narceus annularus*	NC_003343.1
			*Spirobolus*	*Spirobolus bungii*	NC_056899.1
	Spirostreptida	Harpagophoridae	*Thyropygus*	*Thyropygus* sp. DVL-2001	NC_003344.1
		Odontopygidae	*Chaleponcus*	*Chaleponcus netus*	NC_062683.1
			*Prionopetalum*	*Prionopetalum kraepelini*	NC_062688.1
		Spirostreptidae	*Archispirostreptus*	*Archispirostreptus gigas*	NC_062689.1
			*Macrolenostreptus*	*Macrolenostreptus orestes*	NC_062682.1
			*Pseudotibiozus*	*Pseudotibiozus cerasopus*	NC_062681.1
			*Tropostreptus*	*Tropostreptus austerus*	NC_062687.1
				*Tropostreptus droides*	NC_062686.1
				*Tropostreptus hamatus*	MT394521.1
				*Tropostreptus kipunji*	MT394511.1
				*Tropostreptus microcephalus*	NC_062684.1
				*Tropostreptus severus*	NC_062685.1
				*Tropostreptus sigmatospinus*	MT394526.1
Arachnida	Mesostigmata	Ologamasidae	*Stylochyrus*	*Stylochyrus rarior*	CQ927176.2

**Table 2 genes-13-01587-t002:** Mitogenomic organization of *S. bungii*.

Gene	Location		Size	Intergenic	Codon		Stand
Name	From	To	(bp)	Nucleotides	Start	Stop	
rrnS	12	814	803	11			J
trnV	815	873	59				J
rrnL	874	2143	1270				J
trnL1	2165	2227	63	21			J
trnL2	2228	2290	63				J
ND1	2291	3212	922		ATA	T	J
trnP	3213	3275	63				J
ND4L	3277	3558	282	1	ATG	TAG	J
ND4	3552	4893	1342	−7	ATG	T	J
trnH	4894	4956	63				J
ND5	4957	6658	1702		ATT	T	J
trnF	6659	6719	61				J
trnY	6716	6777	62	−4			J
trnQ	6780	6847	68	2			J
trnT	6888	6948	61	40			N
trnS	6953	7016	64	4			N
CYTB	7017	8133	1117		ATG	T	N
ND6	8126	8581	456	−8	ATT	TAA	N
trnE	8582	8642	61				N
trnS	8643	8699	57				N
trnN	8700	8762	63				N
trnR	8762	8823	62	−1			N
trnA	8823	8884	62	−1			N
ND3	8885	9230	346		ATT	T	N
trnG	9231	9293	63				N
COX3	9294	10,071	778		ATG	T	N
ATP6	10,072	10,747	676		ATG	T	N
ATP8	10,741	10,896	156	−7	ATT	TAA	N
trnD	10,897	10,958	62				N
trnK	10,958	11,023	66	−1			N
COX2	11,024	11,701	678		ATG	TAA	N
COX1	11,705	13,234	1530	3	CGA	TAA	N
trnC	13,240	13,302	63	5			J
trnW	13,295	13,356	62	−8			N
ND2	13,357	14,356	1000		ATA	T	N
trnM	14,357	14,419	63				N
trnI	14,420	14,483	64				N

**Table 3 genes-13-01587-t003:** Composition and skewness in the mitochondrial genome of *S. bungii*.

Region	A%	T%	AT-Skew	G%	C%	GG-Skew
Whole mitogenome	26.60	32.62	−0.102	28.44	12.34	0.395
PCGs	24.53	32.22	−0.135	20.46	22.78	−0.054
rRNAs	31.02	35.12	−0.062	22.77	11.10	0.345
tRNAs	34.33	30.69	0.056	19.05	15.93	0.089

## Data Availability

The data that support the findings of this study are openly available in the US National Center for Biotechnology Information (NCBI database) (available online: https://www.ncbi.nlm.nih.gov/nuccore/NC_056899.1, accessed on 22 June 2022).

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
