# Peer review of "The Complete Mitochondrial Genome of Spirobolus bungii (Diplopoda, Spirobolidae): The First Sequence for the Genus Spirobolus"

_genes, 2022, doi:10.3390/genes13091587_

Round 1

Reviewer 1 Report

Dear Authors

Thank you for your submission. Article is overall of interests, but needs to improve the introduction and discussion section. 

Introduction is not very organized, i will suggest to add some more data relevant to the study. 

Discussion is almost absent in the article. I am suggesting to make separate section for it. If authors is interested to combine it with results, please make it coherence with your findings.

Results and M&M section are almost better. 

Improve abstract and conclusion as well. 

Reviewer 2 Report

Dear Editor;

Although the paper written about the complete mitochondrial genome of Spirobolus bungii is based on an original  molecular hypothesis, some items  still need to be proved.

The resolution of Figure 3,4 and 6 should be increased.

Studies carried out in 2020 and beyond should be added to the reference part.

The article is written using a very good and understandable language. Apart from these few minor errors, the article is acceptable.

Round 2

Reviewer 1 Report

Dear Author

No substantial changes has been found in the revised manuscript as suggested.